# Piperitone (*p*-Menth-1-En-3-One): A New Repellent for Tea Shot Hole Borer (Coleoptera: Curculionidae) in Florida Avocado Groves

**DOI:** 10.3390/biom13040656

**Published:** 2023-04-06

**Authors:** Paul E. Kendra, Wayne S. Montgomery, Nurhayat Tabanca, Elena Q. Schnell, Aimé Vázquez, Octavio Menocal, Daniel Carrillo, Kevin R. Cloonan

**Affiliations:** 1Subtropical Horticulture Research Station, United States Department of Agriculture (USDA), Agricultural Research Service (ARS), 13601 Old Cutler Road, Miami, FL 33158, USA; paul.kendra@usda.gov (P.E.K.); wayne.montgomery@usda.gov (W.S.M.); nurhayat.tabanca@usda.gov (N.T.); elena.schnell@usda.gov (E.Q.S.); aime.vazquez@usda.gov (A.V.); 2Tropical Research and Education Center, University of Florida, 18905 SW 280 ST, Homestead, FL 33031, USA; omenocal18@gmail.com (O.M.); dancar@ufl.edu (D.C.)

**Keywords:** ambrosia beetle, *Euwallacea perbrevis*, *Fusarium* dieback, invasive species, *Persea americana*

## Abstract

The tea shot hole borer, *Euwallacea perbrevis*, has been recently established in Florida, USA, where it vectors fungal pathogens that cause *Fusarium* dieback in avocado. Pest monitoring uses a two-component lure containing quercivorol and α-copaene. Incorporation of a repellent into IPM programs may reduce the incidence of dieback in avocado groves, particularly if combined with lures in a push–pull system. This study evaluated piperitone and α-farnesene as potential repellents for *E. perbrevis*, comparing their efficacy to that of verbenone. Replicate 12-week field tests were conducted in commercial avocado groves. Each test compared beetle captures in traps baited with two-component lures versus captures in traps containing lures plus repellent. To complement field trials, Super-Q collections followed by GC analyses were performed to quantify emissions from repellent dispensers field-aged for 12 weeks. Electroantennography (EAG) was also used to measure beetle olfactory response to each repellent. Results indicated that α-farnesene was ineffective; however, piperitone and verbenone were comparable in repellency, achieving 50–70% reduction in captures, with longevity of 10–12 weeks. EAG responses to piperitone and verbenone were equivalent, and significantly greater than response to α-farnesene. Since piperitone is less expensive than verbenone, this study identifies a potential new *E. perbrevis* repellent.

## 1. Introduction

The tea shot hole borer (originally *Euwallacea fornicatus* Eichhoff) (Coleoptera: Curculionidae, Scolytinae, Xyleborini) is an ambrosia beetle native to Southeast Asia, where it has long persisted as a major pest of cultivated tea, *Camellia sinensis* (L.) Kuntze (Theaceae) [1]. However, in recent years it has become evident that there is a complex of invasive beetle species morphologically indistinguishable from tea shot hole borer (collectively *E*. near *fornicatus*). Members of this cryptic complex were recovered in 2012 in Israel from stressed and dying avocado trees, *Persea americana* Miller (Lauraceae) [2]. Infestations have since been detected in avocado groves and native forest trees in the USA, including in California [3] and Florida [4,5], as well as in Mexico [6], South Africa [7], Australia, and other countries [8]. Based on phylogenetic analysis, there are at least five distinct species in the *E.* nr. *fornicatus* complex associated with nine fungal symbionts in the genera *Fusarium*, *Graphium*, and *Acremonium* [4,9,10,11,12]. A recent taxonomic assessment of this beetle complex [13] reassigned the common name “tea shot hole borer” to *E. perbrevis* Schedl, “polyphagous shot hole borer” to *E. fornicatus* Eichhoff, and Kuroshio shot hole borer to *E. kuroshio* Gomez and Hulcr [13].

The species established in South Florida is *E. perbrevis*. Like other ambrosia beetle species [14,15,16,17,18], adult females house conidia of symbiotic fungi within cuticular pouches (mycangia) at the base of their mandibles and inoculate the brood galleries within host trees with these fungi [19]. Larvae and teneral adults feed on the cultivated fungal gardens during development, not the host wood itself. Eleven fungal associates have been recovered from *E. perbrevis* in the avocado production area of Homestead, FL, including *Fusarium* sp. nov., *Fusarium* sp. AF-6, *Fusarium* sp. AF-8, *Graphium euwallaceae*, *Acremonium* sp., *Acremonium murorum*, *Acremonium masseei*, *Elaphocordyceps* sp., and three yeast species [4]. Presence of these fungi results in lesions surrounding beetle galleries and a vascular disease known as *Fusarium* dieback; since dispersing females typically colonize at the base of a branch, infestations usually cause dieback of individual branches, but severe infestation can lead to tree death [2,3,4,11,20,21,22]. Trapping surveys in Florida between 2013 and 2016 recovered *E. perbrevis* from the majority of avocado groves sampled [4,23,24], elevating this species to the status of a major pest of avocado in Miami-Dade County. The beetle was also observed to infest cultivated mango, *Mangifera indica* L. (Anacardiaceae), and soursop, *Annona muricata* L. (Annonaceae) [25]. Additionally, field surveys of natural areas adjacent to avocado groves exhibiting *Fusarium* dieback discovered infestations of *E. perbrevis* in wild tamarind, *Lysiloma latisiliquum* (L.) Bentham (Fabaceae), and *Albizia lebbeck* (L.) Bentham (Fabaceae) [25], suggesting that these hosts may provide an external source of beetles that could infest commercial groves and threaten native forest areas in Florida [20,26].

The prevalence of *E. perbrevis* and *Fusarium* dieback disease in Florida presents further challenges to the $23.6 million dollar avocado industry in the state [27], already threatened by laurel wilt, a lethal vascular disease of avocado and other Lauraceae caused by *Harringtonia lauricola* (T.C. Harr., Fraedrich & Aghayeva) Z. W. de Beer & M. Proctor (Ophiostomatales: Ophiostomataceae) [28,29]. Originally a symbiont unique to the redbay ambrosia beetle *Xyleborus glabratus* Eichhoff [30], *H. lauricola* has transferred laterally to additional species of ambrosia beetle, greatly increasing the pool of vectors [31,32]. Managing *E. perbrevis* may prove difficult, as removing and chipping infested host material is expensive and labor intensive [33], and topical insecticides provide little management for *Euwallacea* sp. pests [34]. One potential strategy for reducing the incidence of *Fusarium* dieback in Florida avocados is incorporating an effective repellent into IPM programs for *E. perbrevis*.

Semiochemical-based attractants are currently used in trapping systems for early detection of pests in the *Euwallacea* nr. *fornicatus* complex [35]. Traps baited with quercivorol lures (which contain a blend of four isomers of *p*-menth-2-en-1-ol) [36] are currently used for *E. fornicatus* and *E. kuroshio* in California [37]. Traps baited with a combination of quercivorol lures and α-copaene lures (which contain a distilled essential oil with 50% (-)-α-copaene content) [24,35,36,38,39] are currently used to optimize detection of *E. perbrevis* in Florida. Previous work has shown that the addition of piperitone (*p*-menth-1-en-3-one) to traps baited with quercivorol significantly reduced *E. fornicatus* and *E. kuroshio* captures in California [37] and *E. fornicatus* captures in Israel [40,41], compared to traps baited with quercivorol alone. In addition, α-farnesene is a repellent for another scolytine beetle, the coffee berry borer, *Hypothenemus hampei* (Ferrari) [42]. The goal of this study was to evaluate piperitone, α-farnesene, and verbenone (the standard repellent for bark beetles, e.g., [43]) as potential new repellents for management of *E. perbrevis* in Florida. This report presents the results of replicated 12-week field tests, electroantennography assays, and chemical analyses of repellent emissions to document the efficacy and longevity of the candidate compounds.

## 2. Materials and Methods

### 2.1. Lures, Repellents, and Traps

Field tests for *E. perbrevis* utilized a two-component attractant consisting of quercivorol and an essential oil enriched in α-copaene [24]. Both were commercially formulated slow-release plastic bubble lures obtained from Synergy Semiochemicals Corp., Delta, BC, Canada. The copaene lures contain 2.0 mL oil in a 2.9 cm diameter bubble (product # 3302); the quercivorol lures contain 97 mg in a 1.2 cm diameter bubble (product #3402). For research purposes, Synergy Semiochemicals also formulated plastic bubble dispensers (2.9 cm diameter) containing 2.0 mL of potential repellent chemicals: piperitone and α-farnesene. These were compared to Synergy’s commercially formulated verbenone dispensers that contain 0.98 g in a 2.9 cm diameter bubble (product #3414).

Trap design (Figure 1) consisted of two white sticky panels (23 cm × 28 cm, Scentry wing trap bottoms; Great Lakes IPM, Vestaburg, MI, USA) suspended back to back from the end of an S-shaped wire hook. Panels were secured along the bottom edge with a binder clip, and the lures/repellent dispensers were clipped to the wire hook above the sticky panels. The final assembly was covered with an inverted clear plastic plate (24 cm diameter) to protect bubble dispensers from rain. Previous field evaluations have shown this sticky trap design to be more effective than conventional Lindgren funnel traps for detection of *E. perbrevis* [35].

### 2.2. Field Evaluation of Piperitone

In total, three 12-week field trapping experiments were conducted in commercial avocado groves in Homestead, Miami-Dade County, Florida, to provide an initial assessment of piperitone repellency. Each grove displayed trees symptomatic of *Fusarium* dieback and had known infestations of *E. perbrevis* [4]. Field test 1 was conducted from 10 October 2017 to 2 January 2018 (site coordinates 25°29.582 N, 80°29.198 W); field test 2 was conducted from 21 December 2017 to 15 March 2018 (site coordinates 25°30.106 N, 80°29.372 W); and field test 3 was conducted from 25 October 2018 to 17 January 2019 (site coordinates 25°35.750 N, 80°29.482 W).

Each test included three treatments: a combination lure (quercivorol and α-copaene), a combination lure plus piperitone, and a non-baited control trap to determine passive captures of inflight females. Tests followed a randomized complete block design, with five replicate blocks in a rectangular grid, and each block consisting of a row of traps. Traps were hung from tree branches in well shaded locations, ~1.5 m above ground with a minimum of 10 m spacing between adjacent traps in a row and 30 m spacing between rows. Tests were serviced every 2 weeks, and at each sampling date, the sticky panels were collected, new panels were deployed, and trap positions were rotated sequentially within each row to minimize positional effects on beetle captures.

Sticky panels collected from the field were taken to the USDA-ARS laboratory (Miami, FL, USA) for processing. All bark and ambrosia beetles were removed from the panels and soaked briefly in a histological clearing agent (Histo-clear 11; National Diagnostics, Atlanta, GA, USA) to remove the adhesive. Specimens were stored in 70% ethanol and later inspected under a dissecting microscope to confirm that only *E. perbrevis* were counted.

### 2.3. Field Comparison of Repellents

Two subsequent field tests were conducted to compare efficacy of piperitone to two known scolytine repellents, verbenone and α-farnesene. Field test 4 was conducted from 7 June to 30 August 2018 at the same grove used for field test 1; field test 5 was conducted from 29 June to 20 September 2018 at the same grove used for test 2. Field tests 4 and 5 included five treatments: a combination lure (quercivorol and α-copaene), a combination lure plus piperitone, a combination lure plus verbenone, a combination lure plus α-farnesene, and a non-baited control trap. Field trapping protocols and sample processing were identical to those described above.

### 2.4. Electroantennography

Beetles for electroantennography (EAG) experiments consisted of female *E. perbrevis* reared from infested avocado branches collected from sites 1 and 2. Branches were held in a black plastic chamber (167 L Brute container with lid; Rubbermaid Commercial Products, Winchester, VA, USA) fitted with screened ventilation holes and a side-mounted wide-mouth mason jar (0.95 L; Ball Corp., Broomfield, CO, USA) to allow for collection of emerging beetles [39]. Each morning, females ≤ 24 h post-emergence were collected for EAG analyses.

Test substrates consisted of piperitone, verbenone, and α-farnesene bubble dispensers (as described above). Sample bottles were prepared by placing a single dispenser into a 250 mL hermetic glass bottle equipped with a lid containing a short thru-hull port (Swagelok, Solon, OH, USA) and silicone septum (Altech, Deerfield, IL, USA). An additional bottle was prepared to provide a standard reference compound, consisting of 5 mL ethanol (95%; Pharmo-Aaper, Brookfield, CT, USA); ethanol has been shown previously to elicit strong EAG responses in female ambrosia beetles [44]. All bottles were sealed and allowed a 2 hr equilibration period for headspace saturation at 24 °C prior to antennal recordings.

Olfactory responses were quantified with a Syntech EAG system and EAG 2000 software (Syntech Original Research Instruments, Hilversum, Netherlands) using methods previously developed for Scolytinae [24,44]. To achieve successful recordings with the minute (~0.5 mm length) antennae of *E. perbrevis* (Figure 2A), a gold-plated 2-pronged probe (Syntech Combi-Probe) was modified by soldering a thin gold wire onto the recording electrode (Figure 2B). A single excised antenna with a small portion of head capsule was mounted onto the indifferent electrode with conductive gel (Spectra 360; Parker Laboratories, Fairfield, NJ, USA), and the gel-coated wire was manipulated to make contact with the underside of the antennal club which lacks olfactory sensilla (Figure 2C). Antennal preparations were placed under a stream of purified, humidified air (400 mL/min), and test substrates (each 1 mL saturated vapor withdrawn from sample bottles) were injected into the airstream using gas-tight syringes (VICI Precision Sampling, Baton Rouge, LA, USA).

EAG responses were recorded from the antennae of 12 females. In each recording session, an antenna was presented first with the ethanol standard, then the repellent samples in random order, then a clean air injection (negative control), followed by a final injection of ethanol. Injections were delivered at 2 min intervals to prevent antennal adaptation. Initial EAG responses were measured in millivolts (peak height of depolarization) and then normalized to percentages relative to the response obtained with ethanol. Normalization with a standard chemical corrects for time-dependent decline in antennal performance, facilitating comparison of relative responses obtained with different substrates [45]. Response obtained with the clean air control was then subtracted from the normalized response to correct for any “pressure shock” caused by the injection volume. The corrected normalized EAG responses were then used for final analyses.

### 2.5. Analysis of Repellent Contents

To determine the chemical content of each repellent, liquid samples (three replicates each) were extracted from the bubble dispensers and analyzed using an Agilent 6890 N GC coupled with a mass spectrometer MS 5975B mass selective detector (Agilent Technologies, Santa Clara, CA, USA). Samples were diluted in methylene chloride in a 1:1000 ratio and 1 μL was injected (PTV splitless, injection temperature of 220 °C, with a solvent delay of 3.75 min). The separation was achieved on a fused silica capillary column coated with a non-polar DB-5 column (5%-phenyl-methylpolysiloxane, 30 m × 0.25 mm × 0.25 μm). Helium was used as a carrier gas at a 1.3 mL/min flow rate. The GC column was held at 60 °C for 1.3 min initially, and then the temperature was programmed from 60 °C, 3 °C/min up to 246 °C [46]. The transfer line and the ion source temperature were 250 °C and 230 °C, respectively. Mass spectra were recorded at 70 eV and full scan mode was *m*/*z* 35–450 with a scan rate of 2.8 scans/sec. Data were analyzed using Mass Hunter B.07.06 software (Agilent Technologies). Headspace volatiles were identified by comparing retention times and linear retention indices (LRI) calculated using the van Den Dool and Kratz equation [47] to a series of *n*-alkanes (C_9_–C_21_, Sigma-Aldrich, St. Louis, MO, USA), library search in Mass-Finder 4 Library [48], Adams Library [46], Flavors and Fragrances of Natural and Synthetic Compounds 3 [49], and Wiley 12/NIST 2020 [50]—an in-house library “SHRS Essential Oil Constituents-DB-5 Column” which was built up from authentic standards and components of known essential oils [51]—and by comparison of the fragmentation patterns with those reported in the literature search [52,53,54]. Eleven compounds were tentatively identified on MS pattern matching scores over 800 from the library results displayed in multiple libraries, and many were further confirmed by their retention indices. In terms of spectral matching scores, searches with scores greater than 900 were considered near perfect, while matching scores greater than 800 were very good and good, respectively, as reported [55]. Four standards were purchased from the following sources: aromadendrene (Cas # 489-39-4), (*E*)-β-farnesene (Cas # 18797-84-8), farnesene, and a mixture of isomers (product number W383902) was purchased from Sigma-Aldrich, St. Louis, MO, USA; and *ar*-curcumene (Cas # 4176-06-1) was purchased from BOC Sciences Shirley, NY, USA. Results are reported as TIC (total ion current) relative percent area.

The enantiomeric composition of piperitone was performed on a Trace GC Ultra (Thermo Scientific, Waltham, MA USA) using an Rt-βDEXse 30 m, 0.32 mm × 0.25 μm capillary column (Restek Corporation, Bellefonte, PA, USA). Helium was used as a carrier gas at 1.2 mL/min. The samples were analyzed with a split ratio of 10:1. The injector and FID (flame ionization detector) temperatures were 225 °C and 230 °C, respectively. The column oven was programmed from 50 °C, held for 2 min, to 100 °C at the rate of 2 °C/min and increased to 220 °C at the rate of 20 °C/min, with a final hold time of 5 min. Samples were diluted with methylene chloride at a 1:1000 ratio and 1 μL of the extract was injected into the GC. Piperitone (CAS: 89-81-6, mixture of enantiomers, predominantly (*R*)-(-)-form) was purchased from TCI America (Portland, OR, USA). The elution order of piperitone was identified by comparing their retention time using co-injection. Triplicate injections were made for each sample. The enantiomeric excess was calculated in the following equation: if the (*R*) isomer is in excess, then the value is obtained by *R*-*S*/*R* + *S* × 100% (for piperitone); if the (*S*) isomer is in excess, then the value is obtained by *S*-*R*/*R* + *S* × 100% (for verbenone).

The enantiomeric composition of verbenone was carried out on an Agilent GC-7890A (Agilent Technologies, Santa Clara, CA, USA). Separation was performed using a Rt-βDEXse 30 m, 0.25 mm × 0.25 μm capillary column (Restek Corporation) with helium as carrier gas at 1.2 mL/min constant flow. The injector and detector temperatures were 225 °C and 230 °C, respectively, with a split ratio of 10:1. The oven temperature was programmed from 40 °C to 220 °C at 4 °C/min and held for 10 min. Samples were diluted with methylene chloride at a 1:1000 ratio and 1 μL of the extract was injected into the GC. (1*S*)-(-)-Verbenone (CAS: 1196-01-6) was purchased from Sigma-Aldrich, St. Louis, MO, USA. The elution order of verbenone was identified by comparing their retention time using co-injection. Triplicate injections were made for each sample.

### 2.6. Volatile Collections and Analysis of Repellent Emissions

Emissions from the commercial repellents (3 replicates of each) were analyzed over a 14-week longevity study. Repellent dispensers were deployed in trap assemblies (as described above but without the sticky panels) hung from trees at the USDA-ARS SHRS (Miami, FL, USA) for the duration of the study. At regular intervals, the traps were brought to the lab and dispensers were removed and sampled separately for volatile emissions using a glass volatile collection chamber (10 cm × 44 cm, Analytical Research Systems, Gainesville, FL, USA) for headspace collection. A purified air stream flowed over the lures at a rate of 1 l/min and was pulled through a 30 mg super-Q adsorbent (Analytical Research Systems, Gainesville, FL, USA) by vacuum for 15 min. Volatiles were then eluted from the filter with 200 μL of methylene chloride (99.8%, J.T. Baker^TM^, Thermo Fisher Scientific Inc., Hampton, NH, USA). An internal standard 5 μg of C-16 (hexadecane CAS: 544-76-3, Sigma-Aldrich) was added for quantification purposes. Subsequently, 1 μL of the collected volatiles was analyzed by an Agilent 7890 GC equipped with a DB-5 column (30 m × 0.25 mm × 0.25 μm, Agilent Technologies). The analysis method used the injector on splitless mode at 225 °C with a 7 psi and helium flow of 1.3 mL/min. The column oven was programmed from 35 °C to 130 °C at the rate of 10 °C/min, and to 220 °C at the rate of 15 °C/min, with a final hold time of 3 min. The detector was an FID at 250 °C. Upon completion of volatile extractions, repellent dispensers were reattached to the trap assemblies, returned to the field, and trap positions were rotated.

### 2.7. Statistical Analysis

One-way analysis of variance (ANOVA) was used to test the effect of treatment on (a) mean captures of *E. perbrevis* (beetles/trap/week) in field tests, and (b) mean antennal responses in EAG experiments. Significant ANOVAs were followed by mean separation with Tukey’s honestly significant difference (HSD) test. When necessary, field capture data were square root (*x* + 0.05)-transformed to stabilize variance prior to ANOVA. Analysis by *t*-test was used for comparisons of factors with two levels, and regression analysis was used to document release rates of volatiles from the piperitone, verbenone, and α-farnesene dispensers. All analyses were performed using SigmaPlot 14.0 (Systat Software Inc., San Jose, CA, USA). Results are presented as mean ± SEM; probability was considered significant at a critical level of α = 0.05.

## 3. Results

### 3.1. Field Evaluation of Piperitone

In field test 1 (Figure 3A), there were significant differences in mean captures of *E. perbrevis* among treatments (*F* = 22.245; df = 212; *p* < 0.001). No beetles were captured with the non-baited control traps, and traps baited with the combination lure plus piperitone captured significantly fewer beetles than traps baited with the lure alone. Presence of piperitone resulted in a mean reduction in captures of 70.86 ± 12.15%. Examination of weekly captures (Figure 3B) indicated that the decrease in captures with piperitone was significant up through week 10 (*t* = 5.308, df = 4, *p* = 0.006), but not significant at week 12 (*t* = 1.089, df = 4, *p* = 0.338).

In field test 2 (Figure 3C), there were significant differences in captures among treatments (*F* = 31.896; df = 212; *p* < 0.001). Numbers of beetles intercepted by the non-baited control trap were significantly lower than captures with either of the baited traps. Traps baited with the lure plus piperitone caught significantly fewer beetles than traps baited with the lure alone. The addition of piperitone decreased captures by 52.40 ± 1.14%. As observed in field test 1, analysis of weekly captures (Figure 3D) indicated that the reduction in captures with piperitone was significant up through week 10 (*t* = 4.982, df = 4, *p* = 0.007), but not week 12 (*t* = 0.563, df = 4, *p* = 0.603). Field trapping also indicated that this site had much higher levels of *E. perbrevis* infestation than the other two avocado groves.

Likewise, in field test 3 (Figure 3E), there were significant differences in beetle captures among treatments (*F* = 17.423; df = 212; *p* < 0.001). Consistent with tests 1 and 2, traps baited with the lure plus piperitone caught significantly fewer beetles than traps baited with lure alone. However, these low numbers with piperitone were not significantly different than those obtained with the non-baited control traps. Presence of piperitone achieved a reduction in captures of 73.05 ± 4.70%, and in this field test the decrease in captures was significant up through week 12 (*t* = 3.277, df = 4, *p* = 0.015) (Figure 3F).

### 3.2. Field Comparison of Repellents

Trap captures in field test 4 were generally lower than those observed in field tests 1–3. In test 4 (Figure 4A), there were significant differences in mean captures of *E. perbrevis* among the five treatments (*F* = 12.668; df = 420; *p* < 0.001). The highest number of captures were obtained with traps baited with the combination lure alone or the lure plus α-farnesene. Traps baited with the lure plus piperitone caught significantly less than the lure alone, but captures were not significantly different from those observed with the lure plus α-farnesene. Traps baited with the lure plus verbenone captured significantly fewer beetles than the lure alone or lure plus α-farnesene, but comparable numbers to those caught with lure plus piperitone or the non-baited control. Addition of piperitone resulted in a mean decrease in captures of 72.98 ± 3.09%, and addition of verbenone in a decrease of 85.16 ± 2.20%. Examination of weekly captures (Figure 4B) indicated that the reduction was significant up through week 12 with both piperitone (*t* = 6.042, df = 8, *p* < 0.001) and verbenone (*t* = 6.810, df = 8, *p* < 0.001).

Similar to field test 4, overall trap captures in field test 5 were lower than those in field tests 1–3. In test 5 (Figure 4C), there were significant differences in mean captures of *E. perbrevis* among treatments (*F* = 6.714; df = 420; *p* = 0.001). Consistent with test 4, traps baited with the combination lure alone or the lure plus α-farnesene captured the highest numbers of beetles. Trap captures with the lure plus piperitone or the lure plus verbenone were significantly lower than both the lure alone or lure plus α-farnesene, and equivalent to the random captures obtained with the non-baited control. Presence of piperitone resulted in a reduction in captures of 84.70 ± 2.73%; presence of verbenone decreased captures by 86.07 ± 1.66%. As observed in field test 4, analysis of weekly captures (Figure 4D) indicated that the reduction in captures with piperitone was significant up through week 12 (*t* = 4.143, df = 8, *p* = 0.003). However, the repellent effect of verbenone was significant at week 10 (*t* = 7.203, df = 8, *p* < 0.001) but not at week 12 (*t* = 2.116, df = 8, *p* = 0.067).

### 3.3. Electroantennography

In the EAG experiment (Figure 5), there were differences in mean olfactory response among treatments (*F* = 8.177; df = 233; *p* = 0.001). Volatile emissions from the piperitone and verbenone dispensers elicited equivalent EAG responses, and both were significantly higher than the response measured with emissions from the α-farnesene dispenser.

### 3.4. Chemical Analysis of Repellent Contents

The GC-MS analysis of the repellent contents indicated that piperitone and verbenone were 100% pure, while the α-farnesene dispenser contained a mixture of 15 sesquiterpene hydrocarbons, predominantly (*E*,*E*)-α-farnesene (42.74%) and (*E*)-β-farnesene (39.52%) (Table 1). Other compounds comprising >1% included (Z)-α-bisabolene, *cis*-cadina-1,4-diene, (*Z*,*E*)-α-farnesene, (*Z*)-γ-bisabolene, (*E*)-γ-bisabolene, d-cadinene, and γ-cadinene. Additional hydrocarbons detected in low amounts included *ar*-curcumene, (*E*)-α-bisabolene, γ-curcumene, β-curcumene, and *trans*-calamenene, with a trace amount of aromadendrene.

Table 2 shows the enantiomeric composition of piperitone and verbenone dispenser contents. The negative enantiomer was predominant in the piperitone dispenser with respect to the enantiomeric excess of 65.86%. The order of elution of the 2 enantiomers was found to be (*R*)-(-)-piperitone at 31.72 min and (*S*)-(+)-piperitone at 31.77 min by co-injection of a commercial piperitone standard. Chiral GC analysis of the verbenone dispenser contents showed that the (S)-(-)-enantiomer was dominant at an enantiomeric excess of 67.64%. The elution order of the enantiomers was found to be (*R*)-(+)-verbenone at 25.32 min and (*S*)-(-)-verbenone at 25.42 min by co-injection of a commercial verbenone standard.

### 3.5. Temporal Analysis of Repellent Emissions

Volatile emissions from the piperitone dispenser (Figure 6A) decreased steadily for 12 weeks after field exposure, at which point the repellent was barely detectable. Piperitone emissions were best fit by the exponential decay model: *y* = −3.32 + 31.85e^(−0.02*x*).^ In contrast, emissions from the verbenone dispenser (Figure 6B) decreased very rapidly during the first week of deployment, and then stabilized at a low but constant rate for the duration of the sampling period (99 days). Verbenone emissions were best fit by the exponential decay model: *y* = 4.47 + 42.38e^(−1.02*x*)^. Emissions from the α-farnesene dispenser (Figure 6C) decreased rapidly during the initial two weeks of field exposure, and then, like verbenone, were maintained at a constant low release rate. Emissions of α-farnesene were best modeled by the exponential decay equation: *y* = 1.42 + 4.77e^(−0.16*x*)^. (Note: Due to a federal government shut down and employee furlough, there was a gap in data collection for emissions from the verbenone and α-farnesene dispensers.)

## 4. Discussion

With the emergence of *E. perbrevis* and *Fusarium* dieback disease in avocado in south Florida, growers need new management tools to protect their groves from this pest. Push–pull as part of an IPM program is one potential option available to avocado growers for managing *E. perbrevis.* This strategy requires the use of repellent stimuli that make the protected crop unattractive, and attractive stimuli that lure the insect away from the location where it can be removed [56]. For example, effective push–pull strategies have been deployed in eastern and southern Africa to manage Pyralid and Noctuid stemborer pests of maize [56,57]. Since quercivorol and α-copaene have already been identified as effective attractants for *E. perbrevis*, the current study tested candidate repellents for this pest.

Female *E. perbrevis* antennae were more responsive to saturated vapor from the piperitone and verbenone dispensers compared to the α-farnesene dispenser. This suggests the presence of a higher number of olfactory receptors for piperitone and verbenone but may also be the result of differences in vapor pressure among the tested compounds. However, the amplitude of EAG responses correlated well with field trials 4 and 5, where α-farnesene was ineffective, but piperitone and verbenone reduced captures by 72–84% and 85–86%, respectively. Our results are consistent with previous work demonstrating that piperitone and verbenone effectively reduced *E. fornicatus* [37,40,41] and *E. kuroshio* [37] captures in quercivorol-baited traps, suggesting that these repellents signal an unsuitable host to *E. perbrevis*, *E. fornicatus*, and *E. kuroshio.* Although α-farnesene did not display repellency in our study, this may have been due to the mixture of compounds in the dispenser; efficacy against *H. hampei* was demonstrated with (*E*,*E*)-α-farnesene [42], which only constituted 43% of the content in the dispensers we tested.

Piperitone and verbenone are monoterpene ketones and oxidation products of plant terpenes. The (*S*)-verbenone isomer is also produced by male Western pine beetles, *Dendroctonus brevicomis* LeConte, when they join and mate with females within host-wood galleries. The addition of (*S*)-verbenone to traps baited with host odors reduced *D. brevicomis* captures [58], likely because it signals an overcrowded tree to conspecifics. Verbenone is also a product of plant degradation [59] and repels some insects that require fresh hosts for colonization [60,61]. Similarly, ethanol (a volatile elevated in stressed and dying trees), when added to quercivorol-baited traps, reduced *E. fornicatus* captures in Israel [41], and reduced captures of both *E. perbrevis* and *X. glabratus* in α-copaene-baited traps in Florida [62]. These invasive species function as primary colonizers of host trees [24,63], and ethanol emissions likely signal poor host quality. Piperitone (*p*-menth-1-en-3-one) is generated by oxidation of quercivorol (*p*-menth-2-en-1-ol) [64]. Due to structural similarities, piperitone may be a competitive inhibitor that blocks quercivorol binding sites on olfactory receptors, thereby reducing attraction [41]. Recently, Kendra et al. [50] documented that the six major symbionts of *E. perbrevis* all emit *trans-p*-menth-2-en-1-ol as a food-based attractant. Therefore, oxidation of this attractant, liberating piperitone, may indicate that symbiont colonies are dying (i.e., piperitone emissions may signal poor food quality, conferring a repellent effect). Since piperitone is just as effective as verbenone, but less expensive, it should be considered as an alternative repellent for *E. perbrevis*. Further research should investigate potential factors affecting the efficacy of piperitone dispensers and their practical application in IPM programs, including the release rate, enantiomeric mixture, placement and number deployed, and economics of developing an optimal field device.

Byers et al. [40] found that release rates from piperitone dispensers deployed with quercivorol-baited traps at 0.52 mg/d and 5.2mg/d reduced *E. fornicatus* captures by 79% and 81%, respectively. In the current study, piperitone and verbenone dispensers were formulated with 2.0 mL of pure compound. Temporal emission analysis showed a steady decrease of piperitone over a 12-week period and a rapid decrease of verbenone during the first 10 days of deployment, followed by a nearly constant level, still detectable at day 99, suggesting that verbenone is an effective *E. perbrevis* repellent at very low levels. Release rate results of piperitone correlated with field trials 1–3 showing a 50–73% reduction in beetle captures for 10–12 weeks. However, dispensers with varying release rates of piperitone may influence both the degree of repellency and field longevity and should be investigated further. Additionally, the most effective deployment time of these lures, i.e., year-round, or only during peak adult *E. perbrevis* activity, should be explored. Chemical analysis showed that the piperitone was 100% pure, with a mixture of 82.93% (*R*)-(-)-piperitone and 17.07% (*S*)-(+)-piperitone. The current work did not test the antennal sensitivity or repellency of the individual isomers, warranting further examination.

Dispenser placement and number may also influence the effectiveness of a repellent. Previous results with *E. fornicatus* in Israel found that piperitone dispensers placed 0.6 m away from quercivorol-baited traps reduced beetle capture by 50%, and that increasing the number of dispensers from 0–3 around quercivorol-baited traps at 0.75 m further reduced trap capture below 50% [41]. However, there was a diminishing efficiency of increasing the number of piperitone dispensers from 3 to 8 at 0.75 m away, suggesting that maximum efficacy could be achieved with three repellent dispensers positioned 0.75 m away from the target tree [41]. All five field trials in the current study deployed a single piperitone dispenser adjacent to the two-component lures, directly above the sticky panel trap. Future work is needed with the piperitone bubble dispenser to determine its effective range of repellency and optimize its field deployment. Further, future work should investigate the deployment rate of piperitone dispensers necessary to achieve protection against *Fusarium* dieback disease in a commercial avocado orchard infested with *E. perbrevis*.

Since piperitone is contained within the formulated bubble dispenser, and not sprayed directly onto trees, toxicity or phytotoxicity issues should not be of concern. Further, piperitone does not emit an offensive odor that could be carried on the wind to communities surrounding avocado groves. The piperitone bubble dispensers used in this study were supplied by Synergy Semiochemicals Corp., which already produces commercial verbenone products, formulated in bubble dispensers and in larger plastic pouches [65]. Therefore, manufacturing costs and logistics should be minimal in transferring the technology for development of piperitone devices. Although there are few large-scale examples of repellents being used in push–pull strategies for managing insect pests [56,66], piperitone should be investigated further for potential utility in managing *E. perbrevis* in Florida avocado groves. In addition, since *E. perbrevis* is the same shot hole borer that impacts the tea industry in India and Sri Lanka [1], efficacy of piperitone should be evaluated on pest populations in Asia as well.

## 5. Conclusions

The tea shot hole borer, *E. perbrevis*, is an invasive ambrosia beetle that vectors fungal pathogens causing *Fusarium* dieback in host trees. In South Florida, USA, avocado is a preferred host. Identification of effective repellents for incorporation into IPM programs may offer avocado growers a tool to protect their groves from beetle attacks and dieback disease. This study examined the efficacy and field longevity of two candidate repellents, piperitone and α-farnesene, as compared to verbenone, the standard bark beetle repellent. The combined results indicated that α-farnesene was ineffective; however, piperitone and verbenone achieved 50–70% reduction in capture of *E. perbrevis* in lure-baited traps. Repellency of both compounds persisted for 10–12 weeks. The lower cost of piperitone makes it an economical alternative. Additional research is needed to investigate factors that may influence the efficacy of piperitone (e.g., release rate, enantiomeric mixtures, dispenser placement) and to determine how to best implement this new repellent in push–pull systems to suppress populations of *E. perbrevis* in Florida avocado groves.

## Figures and Tables

**Figure 1 biomolecules-13-00656-f001:**
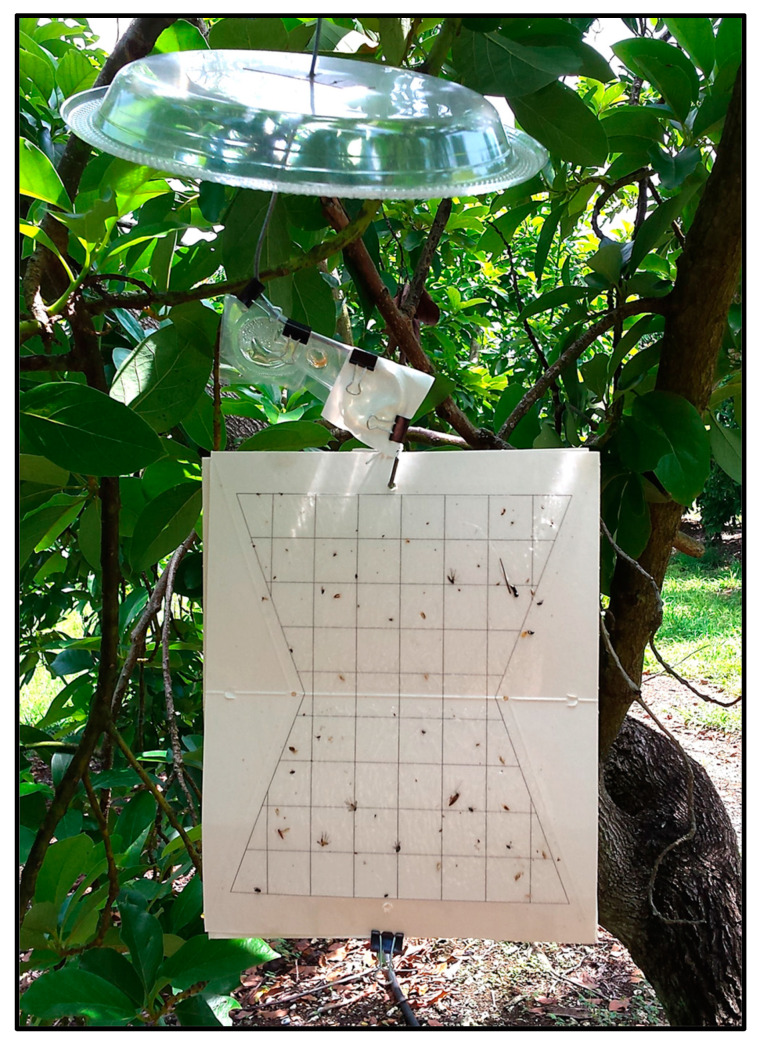
Field trap design. Traps for *Euwallacea perbrevis* were constructed from two opposing sticky panels hung from a wire hook with bubble dispensers clipped above the panels. Dispensers, from left to right, are piperitone, quercivorol, and α-copaene.

**Figure 2 biomolecules-13-00656-f002:**
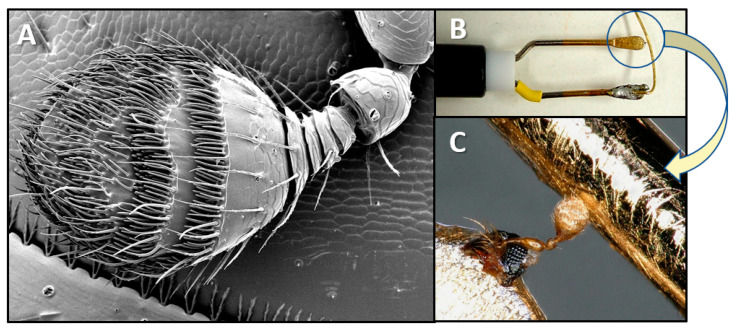
Antenna of female *Euwallacea perbrevis.* (**A**) Scanning electron micrograph; credit: Gary Bauchan, USDA-ARS. (**B**) Two-pronged EAG antennal holder modified with gold wire to accommodate the small antenna of *E. perbrevis*, mounted within the circled region. (**C**) Magnified region from panel B to show antennal preparation used for EAG recordings.

**Figure 3 biomolecules-13-00656-f003:**
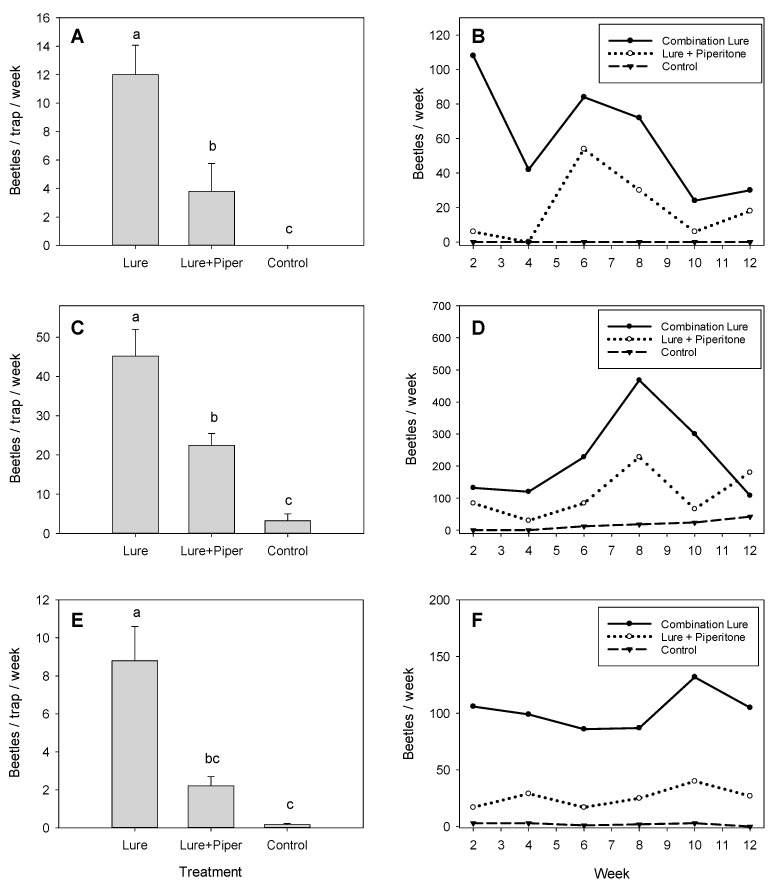
Captures of female *Euwallacea perbrevis* in replicate 12-week field tests evaluating efficacy and longevity of piperitone repellency in three commercial avocado groves, Miami-Dade County, FL, USA. (**A**) Mean (±SE) captures and (**B**) summed weekly captures in field test 1; (**C**) mean (±SE) captures and (**D**) summed weekly captures in field test 2, and (**E**) mean (±SE) captures and (**F**) summed weekly captures in field test 3. Treatments consisted of sticky panel traps baited with a combination lure (quercivorol and α-copaene), traps baited with a combination lure plus piperitone, and non-baited control traps. For panels (**A**,**C**,**E**), bars topped with the same letter are not significantly different (Tukey’s HSD mean separation, *p* < 0.05).

**Figure 4 biomolecules-13-00656-f004:**
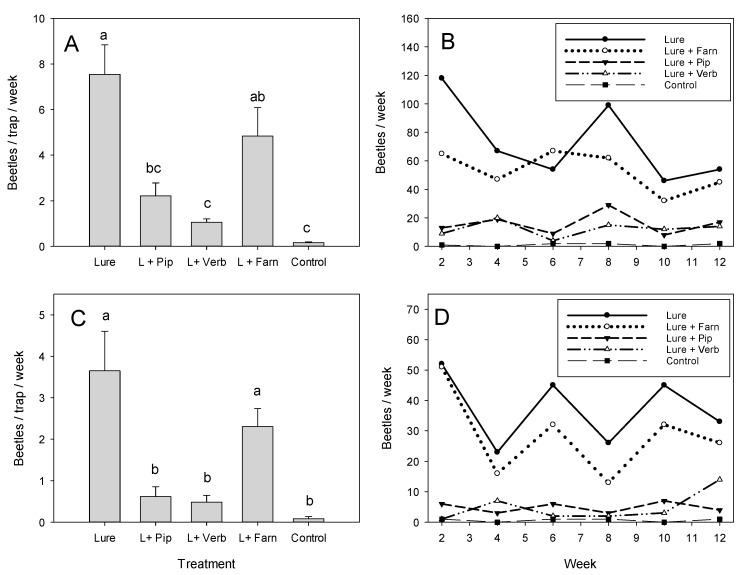
Captures of female *Euwallacea perbrevis* in replicate 12-week field tests comparing repellents deployed in 2 commercial avocado groves, Miami-Dade County, FL, USA. (**A**) Mean (±SE) captures and (**B**) summed weekly captures in field test 4, and (**C**) mean (±SE) captures and (**D**) summed weekly captures in field test 5. Treatments consisted of sticky panel traps baited with a combination lure (quercivorol and α-copaene), traps baited with a combination lure plus piperitone, traps baited with a combination lure plus verbenone, traps baited with a combination lure plus α-farnesene, and non-baited control traps. Bars topped with the same letter are not significantly different (Tukey’s HSD mean separation, *p* < 0.05).

**Figure 5 biomolecules-13-00656-f005:**
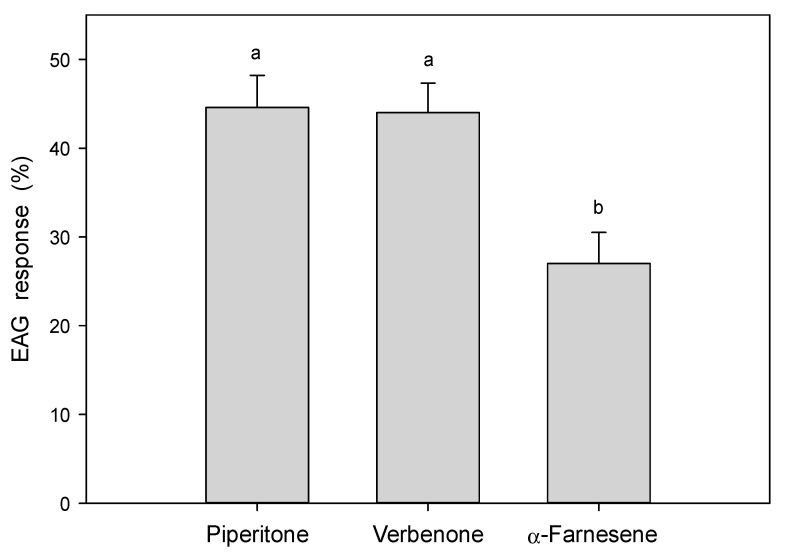
Mean (±SE) electroantennogram responses of female *Euwallacea perbrevis* to volatiles (fixed 1 mL doses of saturated vapor) emitted from bubble formulations of scolytine repellents. EAG responses were normalized (expressed as percentages relative to an ethanol standard [24]). Bars topped with the same letter are not significantly different (Tukey’s HSD mean separation, *p* < 0.05).

**Figure 6 biomolecules-13-00656-f006:**
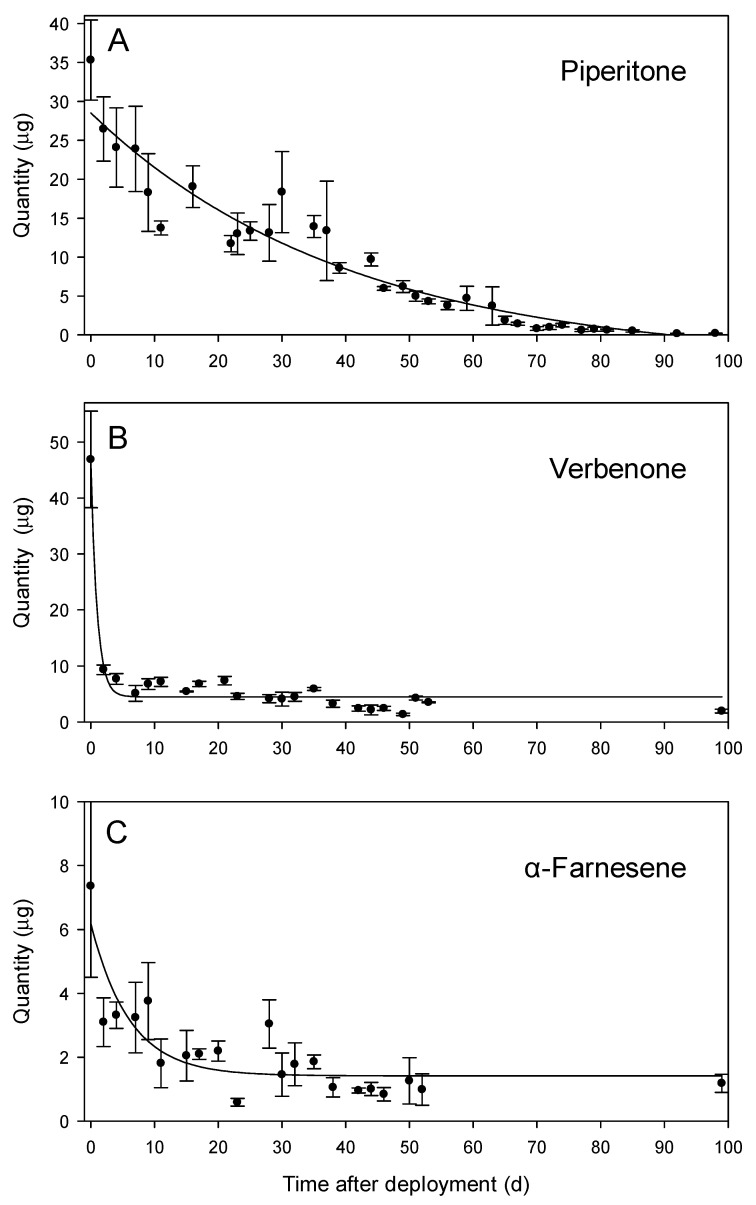
Mean volatile emissions quantified from bubble dispensers containing (**A**) piperitone, (**B**) verbenone, and (**C**) α-farnesene. All dispensers were aged in the field for 99 days and periodically brought into the laboratory for volatile collections and analysis by GC-FID and GC-MS.

**Table 1 biomolecules-13-00656-t001:** Mean (± SE) chemical composition of the α-farnesene dispensers (*n* = 3).

*LKI_exp_	**LKI_lit_	Compounds	%	***Sim.	****IM
1440	1440 ^a^	aromadendrene	0.01 ± 0.00	922	RI, MS, Std
1457	1456 ^a^	*(E*)-β-farnesene	39.52 ± 0.47	908	RI, MS, Std
1480	1480 ^a^	*ar*-curcumene	0.92 ± 0.03	927	RI, MS, Std
1482	1482 ^a^	γ-curcumene	0.43 ± 0.02	921	RI, MS,
1491	1495 ^a^	*cis*-cadina-1,4-diene	1.39 ± 0.03	890	RI, MS,
1494	1499 ^b^	(*Z*,*E*)-α-farnesene	1.30 ± 0.00	875	RI, MS,
1502	1505 ^c^	(*Z*)-α-bisabolene	5.74 ± 0.06	922	RI, MS,
1506	1505 ^a^	(*E*,*E*)-α-farnesene	42.74 ± 0.31	921	RI, MS, Std
1511	1512 ^d^	β-curcumene	0.20 ± 0.02	866	RI, MS,
1513	1513 ^a^	γ-cadinene	1.05 ± 0.02	867	RI, MS,
1515	1515 ^a^	(*Z*)-γ-bisabolene	3.35 ± 0.05	894	RI, MS,
1521	1522 ^a^	*trans*-calamenene	0.12 ± 0.01	849	RI, MS,
1522	1523 ^a^	δ-cadinene	1.10 ± 0.02	888	RI, MS,
1530	1531 ^a^	(*E*)-γ-bisabolene	1.29 ± 0.02	874	RI, MS,
1545	1547 ^a^	(*E*)-α-bisabolene	0.85 ± 0.02	913	RI, MS,
		Total	100 ± 0.00		

*LKI_exp_: relative retention indices determined on the experimental DB-5 GC column against *n*-alkanes; **LKI_lit_: relative retention indices from the literature (^a^ [46], ^b^ [52], ^c^ [53], ^d^ [54]); ***Sim.: similarity scores to the reference spectrum. ****IM: identification method; RI: retention index; MS: computer matching of the mass spectra with those of the Adams Library [46], MassFinder [48], FFNSC3 [49], and Wiley 12/NIST2020 [50] libraries and comparison with data from the literature [52,53,54]; Std: standards compounds were purchased.

**Table 2 biomolecules-13-00656-t002:** Mean (± SE) enantiomeric composition of piperitone and verbenone in repellent dispensers and standard reference samples (*n* = 3) using an Rt-βDEXse column.

Samples	(*R*)-(-)- Piperitone	(*S*)-(+)- Piperitone	(*R*)-(+)- Verbenone	(*S*)-(-)- Verbenone
Piperitone dispenser	82.93 ± 0.21	17.07 ± 0.21		
Piperitone standard	87.66 ± 0.10	12.34 ± 0.10		
Verbenone dispenser			16.18 ± 0.01	83.82 ± 0.01
Verbenone standard			22.68 ± 0.12	77.32 ± 0.12

## Data Availability

The data are available from the authors upon request.

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
