# Peer review of "Piperitone (p-Menth-1-En-3-One): A New Repellent for Tea Shot Hole Borer (Coleoptera: Curculionidae) in Florida Avocado Groves"

_biomolecules, 2023, doi:10.3390/biom13040656_

Round 1

Reviewer 1 Report

The manuscript, Piperitone (p-menth-1-en-3-one): A new repellent for tea shot hole borer (Coleoptera: Curculionidae) in Florida avocado, by Kendra et al described good efficiencies of two potential repellents. The concept was, however, previously described, and the authors undertook a systematic study on the potential of the two compounds as repellents providing a step forward to developing a repellent system. The study design, methodology, and conclusion drawn from the results are scientifically sound. One question I want to ask is if it is possible to measure repelling effect itself, rather than measuring a decrease in lure traps. Nonetheless, it has merit to be published in Biomolecules.

Couple of minor comments:

L214, change ‘246 áµ’C, 3 áµ’C/min up to 246 áµ’C’ to ‘246 áµ’C at the rate of 3 áµ’C/min’.

Table 2 summarises enantiomeric distribution of piperitone and verbenone, but the 65.86% of enantiomeric excess of (R)-(-)- isomer is stated in the text. Clarify where this enantiomeric excess comes from. Clarify the same issue with verbenone.

Reviewer 2 Report

This study identifies an economical new repellent, which is very helpful for integrated for IPM of E. perbrevis. The authors utilized field trapping, electroantennography, and analysis of repel-lent emissions to examine the efficacy and field longevity of two candidate repellents, piperitone and α-farnesene, Piperitone and verbenone achieved 50-70% reduction in capture of E. perbrevis in lure-baited traps, with longevity 10-12 weeks. Since piperitone is less expensive than verbenone, the lower cost of piperitone makes it an economical alternative. The methods are correct and the conclusions are supported by the results. Some points need to be addressed by the authors before publication of this manuscript.

1. “Lure+Verb” significantly decreased captures in replicate 12-wk field tests, but the volatile emissions of verbenone quickly decreased to a low level 10 days ago in the Figure 6, please analyze in the discussion that Lure+Verb can still have a good ability to reduce the captures of female Euwallacea perbrevis in replicate 12-week field tests.

2. Please add data points in the Figure3(A,C,E) and Figure4.

          3. In field test 4 and 5, what are the captures of Lure+Pip, Lure+Verb and Lure+Farn at different weeks, and are there obvious differences?

Reviewer 3 Report

This manuscript describes some experiments testing piperitone and farnesene as repellents for an invasive bark beetle, comparing it to the generic bark beetle repellent verbenone. The manuscript is well written, the experiments appear to have been done competently, and the results are clearly described. My sole major concern is that the authors are reading too much into their interpretation of the results, and the downstream implications for their results, i.e., they have stated or implied (e.g. L 27) that piperitone is an economical new repellent for IPM. However, so far, all they have demonstrated is a very preliminary step, i.e., that piperitone can decrease captures of beetles in traps baited with a host plant type attractant. That is a very long way from having an effective and logistically feasible repellent-based system for management of the beetle. In short, they should be a lot more conservative in their projections for this compound at this very early stage. In the discussion, they have superficially mentioned a few of the hurdles that must be overcome in order to develop a viable, economical and field-scale system, but a more explicit and thoughtful discussion of those and other potential hurdles, and particularly, an explicit statement re the potential toxicity and phytotoxicity, or lack thereof, would be very useful. If piperitone has a known toxicity profile, then chances of using it become slimmer. Conversely, if it is essentially benign, then this is a big selling point.

Specific comments as follows:

1.       L 69: Is it really true that the avocado industry is worth only about $24 million in Florida? I had just expected that it would be a much larger industry/net worth, akin to California (?) for example.

2.       L 77-79, and/or in the discussion: are there any cropping systems, anywhere, in which repellents form an effective component of IPM over substantial portions of the total crop acreage? If yes, then the authors might want to cite some examples, to show that there is at least some precedent for such an approach.

3.       L 171 and elsewhere, recording electrode rather than the different electrode

4.       L 210-211, repetition in the sentence. Same on L 214, 360.

5.       L 215, it seems rather odd that the transfer line temp (200C) was maintained about 50 degrees below the final oven temp of 246C, i.e., the transfer line would represent a cool spot, potentially slowing down and broadening out later eluting peaks. Here, for the analysis of the volatile monoterpenoids, it probably would not matter, but it will certainly matter for analyses of less volatile compounds in other analyses.

6.       L 227 and elsewhere, change enantiomeric distribution to enantiomeric composition.

7.       In the results and/or discussion, the authors might want to point out that the trap catches in expts 4 and 5 were low, maximums of less than 8 or less than 4 beetles per week respectively, and so these results should be interpreted cautiously, i.e., they are probably not nearly as robust as the results in expt 2.

8.       Fig 5 caption, explicitly state that the Y axis percent values are in relation to responses to ethanol.

9.       Table 1: explicitly state which of these identifications were confirmed by matches with authentic standards by e.g., bolding the matched ones and stating that in the caption, and which ones are still only tentative from database matches, particularly as the match qualities are not stated. In particular, sesquiterpenes are notoriously difficult to identify by mass spectra alone, and matching KIs helps but is still not quite definitive. Matches with database spectra can only be as good as the compounds which are in the database, i.e., if the compound is not in the database, it will still give you a best match, even though that match is incorrect.

10.   L 366, analysis of the verbenone dispenser contents, or volatiles, not the dispenser itself.

11.   L 401: here, and with the use of ethanol as the reference standard, the authors have not accounted for the differences in doses delivered to the antennae, i.e., they used 1 ml of saturated vapor of each compound, but the numbers of molecules delivered to the antennae would be determined by the compounds’ vapor pressures, which likely differed by orders of magnitude. Thus, the authors might want to add a caveat to this effect.

12.   L 421, the authors suggest that verbenone and piperitone may signal similar information about a heavily colonized host. What basis do the authors have for making this statement? Is there evidence that their bark beetle species produces verbenone to regulate attack densities, or even more pertinent, that their species produces piperitone to regulate attack densities? If not, they should phrase this a lot more carefully. I.e., you cannot imply that a beetle regulates its host density using a compound it is not known to produce!

13.   L 428-449 should be revised and expanded to provide a better overview of the possibilities of developing piperitone as an effective IPM tool for protecting avocados as a crop. The authors have cited precedents where people have done similar experiments to theirs, i.e., decreasing trap catch, which is a first preliminary indication that piperitone could possibly be developed as a large scale repellent. However, following that, rather than vague statements about how more research is needed to determine dose, field longevity, etc., it would be most useful to the reader to provide a thoughtful description of the main hurdles that need to be overcome, which might include:

a.        toxicity and phytotoxicity, may not be a problem

b.      potential blowback from the public, i.e., piperitone is volatile, and if it has a distinctive smell that carried on the wind to houses or communities surrounding avocado groves, this could be an issue

c.       the economics and logistics of formulating, loading, and deploying dispensers in sufficient densities to have a chance of being effective, possibly by relating what is likely to be needed in this system to other systems in which bubble caps are used, to give some idea of doses and densities needed, and from there costs of materials and labor.

d.      the time period over which the repellents would need to be deployed, i.e., year round, or only during certain periods of the year when the beetles are moving to new hosts.

e.      an explicit description of the insect vector and host system, i.e., the authors were getting 70-80 percent trap shutdown, and for the sake of argument, if you assumed that you would get the same level of deterrence if the piperitone dispensers were deployed thoughout the crop, would this result in similar levels of crop protection, i.e., 70-80% less infection? And specifically, can a single beetle transmit enough fungus to kill a tree, or does it take multiple beetles before the tree is overwhelmed? This alone would be a crucial piece of data that could determine the success or failure of a repellent tool. That is, if insecticide sprays or other control measures would still be needed even after deploying the repellents, then using repellents might simply add to control costs.

In short, as it is, the takehome message that the reader gets is that a lot more research needs to be done. This is not very helpful. What would be much more useful for the reader would be a well thought out and balanced discussion of the major points that need to be considered and addressed if this is to become a viable IPM tool.

14.   L 455-456, electroantennography did not really contribute to testing the efficacy or field longevity?

Round 2

Reviewer 3 Report

The authors have made a good faith effort to address my major concern, and most of the other points raised. A couple of final points that they should consider, as below. These can be easily dealt with, i.e., it is not necessary for any further changes to be reviewed again.

Original point:

This manuscript describes some experiments testing piperitone and farnesene as repellents for
an invasive bark beetle, comparing it to the generic bark beetle repellent verbenone. The
manuscript is well written, the experiments appear to have been done competently, and the
results are clearly described. My sole major concern is that the authors are reading too much
into their interpretation of the results, and the downstream implications for their results, i.e.,
they have stated or implied (e.g. L 27) that piperitone is an economical new repellent for IPM.
However, so far, all they have demonstrated is a very preliminary step, i.e., that piperitone can
decrease captures of beetles in traps baited with a host plant type attractant. That is a very long
way from having an effective and logistically feasible repellent-based system for management
of the beetle. In short, they should be a lot more conservative in their projections for this
compound at this very early stage. In the discussion, they have superficially mentioned a few of
the hurdles that must be overcome in order to develop a viable, economical and field-scale
system, but a more explicit and thoughtful discussion of those and other potential hurdles, and
particularly, an explicit statement re the potential toxicity and phytotoxicity, or lack thereof,
would be very useful. If piperitone has a known toxicity profile, then chances of using it become
slimmer. Conversely, if it is essentially benign, then this is a big selling point.

Rebuttal:

We thank the reviewer for the time and thought invested in this review. Our intent was to
provide a first report of piperitone as a repellent chemical for E. perbrevis, which we feel is
appropriate for the journal ‘Biomolecules’. This is not an article for an economic entomology
journal, and we apologize if there has been a misinterpretation that piperitone is ‘field ready’ as
an IPM component. Throughout the discussion, we have used the terms ‘potential’ and
‘candidate repellents’ and we clearly acknowledged that further research is needed to
determine range of repellency, factors that affect efficacy, and optimization of a field device.
The revised text, which now includes an additional paragraph in the discussion, incorporates
new considerations suggested by the reviewer, thereby improving the manuscript.

For further consideration:

è The authors also need to tone down the abstract, the last sentence of which still states “Since piperitone is less expensive than verbenone, this study identifies an economical new repellent for IPM of E. perbrevis in Florida”, which implies that piperitone is indeed ready for deployment

Original point 9. Table 1: explicitly state which of these identifications were confirmed by matches with
authentic standards by e.g., bolding the matched ones and stating that in the caption, and
which ones are still only tentative from database matches, particularly as the match qualities
are not stated. In particular, sesquiterpenes are notoriously difficult to identify by mass spectra
alone, and matching KIs helps but is still not quite definitive. Matches with database spectra can
only be as good as the compounds which are in the database, i.e., if the compound is not in the
database, it will still give you a best match, even though that match is incorrect.

Rebuttal: This has been addressed in the Materials and Methods in ‘Analysis of Repellent -->Contents’ with the sentence, “Eleven compounds were tentatively identified with the similarity over 800. Fourstandards were purchased from the following sources: aromadendrene (Cas # 489-39-4), (E)-b-farnesene (Cas # 18797-84-8), farnesene, mixture of isomers (product number W383902) fromSigma-Aldrich, St. Louis, MO, USA, and ar-curcumene (Cas 4176-06-1) was purchased BOC
Sciences Shirley, NY, USA.”

è Clarify what is meant by a similarity over 800, i.e., this may be unclear to those who are not familiar with using MS database searches. Also, I am somewhat surprised that the authors would consider a similarity match of 800 as a good match, particularly for compounds like sesquiterpenes, of which there are hundreds of structures, and where their mass spectra can be very similar.

Original point 12. L 421, the authors suggest that verbenone and piperitone may signal similar information about a heavily colonized host. What basis do the authors have for making this statement? Is there evidence that their bark beetle species produces verbenone to regulate attack densities, or
even more pertinent, that their species produces piperitone to regulate attack densities? If not,
they should phrase this a lot more carefully. I.e., you cannot imply that a beetle regulates its
host density using a compound it is not known to produce!

Rebuttal:

This issue has been addressed in the third paragraph of the Discussion section with the
following sentences, “Verbenone is also a product of plant degradation [58] and repels some
insects that require fresh hosts for colonization [59,60]. Similarly, ethanol (a volatile elevated in
stressed and dying trees) when added to quercivorol-baited traps reduced E. fornicatus captures
in Israel [41], and reduced captures of both E. perbrevis and X. glabratus in α-copaene-baited
traps in Florida [61]. These invasive species function as primary colonizers of host trees [24,62],
and ethanol emissions likely signal poor host quality. Verbenone and piperitone may signal
similar information about an unsuitable host to dispersing E. fornicatus and E. perbrevis.”

è  The authors have not addressed the original question, i.e., what basis do they have for stating that verbenone and piperitone may signal similar information about colonized hosts? They have discussed how verbenone and ethanol might signal unsuitable hosts, but why should piperitone act in similar fashion, when it is not known to be produced by either the beetles or the host trees, as far as I can tell. I.e., of all the possible compounds that could have been tested, what was the justification or underlying rationale for testing piperitone?

Original point 13e:

e. an explicit description of the insect vector and host system, i.e., the authors were getting 70-
80 percent trap shutdown, and for the sake of argument, if you assumed that you would get the
same level of deterrence if the piperitone dispensers were deployed thoughout the crop, would
this result in similar levels of crop protection, i.e., 70-80% less infection? And specifically, can a
single beetle transmit enough fungus to kill a tree, or does it take multiple beetles before the
tree is overwhelmed? This alone would be a crucial piece of data that could determine the
success or failure of a repellent tool. That is, if insecticide sprays or other control measures
would still be needed even after deploying the repellents, then using repellents might simply
add to control costs.

Rebuttal:

The authors believe that sufficient background information was already provided in the
introduction, including details regarding branch dieback and tree mortality (and ample
references should a reader desire addition information):
‘The species established in South Florida is E. perbrevis. Like other xyleborine ambrosia beetle
species [14-18], adult females house conidia of symbiotic fungi within cuticular pouches etc.

è The authors have dodged the question and not provided a crucial piece of information that would enter into a decision as to whether a repellent strategy might have a chance of success. That is, is infestation by a single beetle and its fungus likely to doom a tree, so that 70-80% repellence would likely be insufficient to protect an orchard? They state that infestations ususally cause branch dieback, possibly implying that diligent pruning might save the tree, and that severe infestation can lead to tree death, implying that attack by a single beetle or a few beetles will not doom the tree as long as it is pruned. I would interpret both of these statements as supporting their case that repellents might work, and they might want to suggest this, rather than leaving it up in the air.
